# Ultrasonographic and Three-Dimensional Analyses at the Glabella and Radix of the Nose for Botulinum Neurotoxin Injection Procedures into the Procerus Muscle

**DOI:** 10.3390/toxins11100560

**Published:** 2019-09-24

**Authors:** Younghoon Cho, Hyung-Jin Lee, Kang-Woo Lee, Kyu-Lim Lee, Jae Seung Kang, Hee-Jin Kim

**Affiliations:** 1Medical R&D Center, Bodyfriend, 163 Yangjaecheon-ro, Gangnam-gu, Seoul 06302, Korea; snake4th@gmail.com; 2Division in Anatomy and Developmental Biology, Department of Oral Biology, Human Identification Research Institute, BK21 PLUS Project, Yonsei University College of Dentistry, 50-1 Yonsei-ro, Seodaemun-gu, Seoul 03722, Korea; LEEHJ221@yuhs.ac (H.-J.L.); leekw@yuhs.ac (K.-W.L.); kyulimlee@yuhs.ac (K.-L.L.); 3Department of Anatomy and Tumor Immunity Medical Research Center, College of Medicine, Seoul National University, 103 Daehak-ro, Jongno-gu, Seoul 03080, Korea; genius29@snu.ac.kr; 4Department of Materials Science & Engineering, College of Engineering, Yonsei University, 50 Yonsei-ro, Seodaemun-gu, Seoul 03722, Korea

**Keywords:** M. procerus, glabella, sellion, radix of the nose, botulinum neurotoxin injection, facial rejuvenation procedures, ultrasonographic imaging, 3D scanning imaging, glabellar transverse line

## Abstract

Botulinum neurotoxin (BoNT) injections are widely used for facial rejuvenation procedures, and the procerus muscle is a major target in cases of glabellar transverse lines or rhytids. Although there have been many cadaveric studies of the procerus, its depth and thickness have not been investigated thoroughly. The aim of this study was to measure the depth and thickness of the procerus and identify the location of the intercanthal vein using ultrasonographic (US) imaging and the three-dimensional scanning method, which is needed to know to avoid side effects during BoNT injections. The morphology of the procerus was classified into two types based on the US images obtained at the glabella. The procerus was located deeper below the skin surface at the glabella than the sellion (3.8 ± 0.7 mm versus 2.7 ± 0.6 mm). The width of the procerus in US images increased from the sellion (10.9 ± 0.2 mm) to the glabella (14.5 ± 4.6 mm), whereas its thickness decreased (from 1.6 ± 0.6 mm to 1.1 ± 0.5 mm). The intercanthal vein was located 5.1 ± 4.0 mm superior to the sellion and 3.0 ± 0.6 mm below the skin’s surface. The present findings provide anatomical knowledge as well as the reference location information for use when injecting BoNT into the procerus.

## 1. Introduction

Botulinum neurotoxin (BoNT) injections into the procerus muscle are widely used for facial rejuvenation to decrease wrinkles in cases of glabellar transverse lines or rhytids [1,2,3]. To maximize the efficacy and minimize the side effects associated with BoNT injections, it is important to know the exact anatomical relationship between the procerus and its adjacent muscles and vessels.

The procerus (M. procerus, pyramidalis nasi, depressor glabellae) is a small muscle located at the glabella and upper part of the nose. It arises from the fascia, which is located near the junction of the nasal bones, and the upper lateral nasal cartilage [4]. Previous cadaveric studies found that it consisted of superficial and deep layers and extended to the frontalis superiorly and the nasalis inferiorly [5]. This muscle makes a horizontal line on the radix of the nose below the glabella by pulling the medial side of the eyebrow down. Variations in its insertion points into the glabella and forehead result in various frowning patterns among individuals [6,7,8]. The procerus is known to be innervated by the buccal branch of the facial nerve [9]. Its blood supply is from branches of the facial artery [4]. In the nasoglabellar region, the radix of the nose is regarded as a potential danger zone of the face due to the presence of a venous structure called the intercanthal vein (ICV), which comprises a subcutaneous layer superficial to the procerus [10,11]. In the radix of the nose, ICV communicates with and connects angular veins bilaterally. [10,11]. Due to its valveless structure and connection to the cavernous sinus of the midface vein, the venous flow of ICV and the angular vein can propagate infection intracranially or cavernous sinus thrombosis, and the ICV should be considered [12,13,14].

Despite many cadaveric studies of the procerus muscle, its depth and thickness have not been investigated thoroughly. The thinness of the skin and subcutaneous tissues in the nasoglabellar region, where the procerus is located [15], make it important to know the precise depth of this muscle to ensure effective and safe BoNT injection procedures. Similarly, the depth and accurate location of the ICV should be considered in order to prevent intravascular injections of BoNT.

US-guided examinations allow BoNT injections to be performed safely and more effectively than blind injections [16,17]. However, to the best of our knowledge, no previous study has investigated the US-related anatomy of the procerus. Therefore, we aimed to determine the depth and thickness of the procerus using US imaging, and the location of the ICV in order to improve the effectiveness and avoid side effects during BoNT injection procedures.

## 2. Results

Three-dimensional (3D) scanning images revealed the procerus at the glabella and sellion. At the glabella, US images also revealed the procerus and the ICV at the glabella and sellion (Figure 1A). At the glabella, the procerus was located at depths of 4.1 ± 1.3 mm and 3.8 ± 0.7 mm in 3D scanning images and US images, respectively (no significant difference, Table 1). At the sellion, it was located 3.0 ± 1.1 mm and 2.7 ± 0.6 mm below the skin surface in 3D scanning images and US images, respectively (no significant difference, Table 1). The procerus was located deeper below the skin surface at the glabella than the sellion (*p* < 0.001). The width and thickness of the procerus were 14.5 ± 4.6 mm and 1.1 ± 0.5 mm, respectively, at the glabella, and 10.9 ± 2.5 mm and 1.6 ± 0.5 mm at the sellion, indicating that the procerus became significantly thinner and wider from the sellion to the glabella (*p* = 0.006 and *p* < 0.001, respectively).

The morphology of the procerus was classified into two types based in the glabellar US images (Figure 2). Type I procerus muscle was observed clearly and could be easily distinguished from other adjacent facial expression muscles in transverse-view US images (55 of 69 cases, 79.7%) (Figure 2A). In 20.3% of the volunteers the procerus was classified as type II, in which it was not observed at the midline at the glabella and was divided into two lateral portions (Figure 2B). Instead, a thin band-like structure which is characterized as isoechoic with muscle was shown at the midline. The width of the procerus could not be measured in part of type I and type II because its lateral border was not present beyond the US view and it was not observed at the midline of the glabella, respectively. The inability to observe the procerus at the midline at the glabella in type II also meant that its thickness could not be measured. However, the width and thickness of the procerus could be measured at the sellion in all volunteers. The thickness of the procerus and depth of the muscle from the skin surface were presented (Table 2). At the sellion, type I procerus muscles were thicker than type II muscles (1.6 ± 0.6 mm versus 1.3 ± 0.4 mm, *p* = 0.044).

The ICV was observed in the longitudinal view US images obtained between the glabella and sellion in 30 of 69 cases (43.5%, Figure 1). It was located 5.1 ± 4.0 mm superior to the sellion and 3.0 ± 0.6 mm below the skin surface, and located inferior to the sellion in 2 of 30 cases (6.7%). The ICV was mostly distributed between the glabella and sellion, and appeared superficial to the procerus (Figure 3).

## 3. Discussion

Many previous cadaveric studies have investigated the procerus muscle [5,9,10,11], but the depth and thickness of the muscle have not been measured accurately. In contrast to previous studies, we aimed to determine the depth and thickness of the procerus and the ICV using US imaging in healthy subjects as well as using 3D scanning images of cadavers to provide clinical anatomical information when injecting BoNT at the glabella and radix of the nose. Based on the transverse-view US images obtained at the glabella, the type of procerus was newly classified based on the distribution pattern at the glabella.

Hyperactive procerus and corrugator muscles have been targeted in cases of glabellar transverse lines or rhytids [18,19]. A comparative, randomized, double-blind study found that BoNT injections were effective against glabellar frown lines, but also adverse effects were demonstrated [2]. Adverse effects, including headache, temporary blepharoptosis, and injection site problems such as pain, erythema, skin reaction, bruising, and edema, occur in more than 10% of subjects receiving BoNT injections at the glabella [3,20]. Moreover, severe and intractable headaches occurring after BoNT injections at the glabella in five cases (1%) had been reported [21]. This implies that exact targeting and dosing is needed when injecting into the procerus. If the clinician targets the wrong muscles, muscle weakness or blepharoptosis also occur or the effect of the treatment is less than expected.

The ICV is mostly located above the sellion, but its distribution varies considerably between the glabella and the sellion. The ICV was observed in more than 70% of cases in a cadaveric study [10]. However, in our longitudinal view US images the ICV was found in only 43.5% of the subjects from the glabella to the sellion. This difference is likely due to the limited resolution of the US imaging device, pressure on the skin surface during US scanning, or reduced blood flow of the ICV in some subjects who received US scanning. Although US-guided BoNT injections have been demonstrated to be more effective and safer [16,22], the clinician always needs to be aware of the presence of the ICV in order to avoid bruising and intravascular injections, because this vein might be invisible in US images. Because the common points for BoNT injections into the procerus are just above the brow in the midpupillary line between the glabella and radix of the nose [23], the ICV should be considered.

The distribution pattern of the procerus at the glabella could be categorized into two types. For about four-fifths of the subjects the procerus was one portion without dehiscence or division into two parts. However, no muscle fiber was present at the midline of the glabella in one-fifth of the subjects, and so BoNT injections into the procerus in this type might target subcutaneous tissues or the periosteum rather than the muscle. Although this type of procerus is thinner, the glabellar transverse line might be expected in fewer cases; clarifying this requires further investigations. Corrugator supercilli was shown slightly deeper below the procerus, therefore BoNT injection into corrugator supercilli should be considered simultaneously or not in accordance with the purpose of BoNT injection procedures.

We compared the depth of the procerus between US images and 3D-scanning images of cadavers. Because these two methods have been validated previously [15,24,25], we used both modalities simultaneously in this study in order to increase the reliability. The depth of the procerus could be measured in 3D scanning images without any need to apply pressure; however, an important limitation was that the 3D scanning was performed on old-aged cadavers in the supine position. US imaging has the advantage of being able to be performed during actual BoNT injections. A US imaging examination was also performed on a young living subject in order to identify the vascular architecture. It was therefore considered that comparing these two complementary methods provide more reliable measurements of the depth of the procerus.

## 4. Conclusions

Increasing the effectiveness and reducing the adverse effects when injecting BoNT into the procerus require knowledge of its thickness and depth. Moreover, the clinician needs to be aware of location of the ICV—which is located between the glabella or sellion region superficially from the procerus—in order to prevent intravascular injections. Also, the distribution pattern of the procerus at the glabella should be considered, since it influences the effectiveness of BoNT injections. Our findings provide anatomical knowledge about the procerus, as well as reference location information for use when injecting BoNT into the procerus.

## 5. Materials and Methods

### 5.1. 3D Scanning Using a Structured-Light Scanner for Analyzing the Depth of the Procerus

The depths of the procerus at the glabella and sellion were determined by scanning each of 30 adult Korean and 15 adult Thai embalmed cadavers (27 males, 18 females; mean age, 75.6 years) using the following steps with a structured-light 3D scanner (Morpheus3D^®^, Morpheus Company, Yongin, Korea). All of the cadavers were scanned from the front and bilaterally from each side to obtain 3D images, and the three images obtained from each cadaver were merged using Morpheus dental solution (MDS) software 3.0 (version 3.0, Morpheus Company, Yongin, Korea, 2018). The skin and subcutaneous layer were then removed on the left half of the face, including the midline. The dissected face was scanned again with the same process. The two reconstructed 3D images were superimposed based on the undissected side of the face and analyzed by the MDS program (Figure 4). The difference between the two 3D images indicated the depth of each facial muscle below the skin surface. Points at the glabella and sellion were projected perpendicularly from the 3D scanned image of the undissected cadaver onto the 3D scanned image of the dissected cadaver, and the distances between corresponding points were measured using the MDS program to automatically determine the muscle depth below the surface of the face.

The glabella was defined as the most protruding midline point on the smooth part of the forehead above and between the eyebrows. The sellion was defined as the deepest midline point between the forehead and nose. The same definitions also applied to US analyses.

### 5.2. US Analyses of the Procerus and the ICV

US images of 69 Korean volunteers (44 males, 25 females; mean age, 24.1 years) were obtained using a real-time two-dimensional B-mode US device with a high-frequency hockey-stick-type linear transducer (E-Cube 15, IO8-17X, ALPINION MEDICAL SYSTEMS, Seoul, Korea) operating at 15 MHz. None of subjects had received non-invasive or surgical procedures in the forehead region. The glabella and sellion on the face were selected for observations, and landmarks were marked on the skin surface using a waterproof pen. The transducer was positioned just above the skin surface at each landmark, with a US gel applied thickly in order to minimize the pressure to the soft tissue of the face.

US scanning was performed transversely or longitudinally according to the purposes of the observation. In the transverse view, the type, thickness, depth, and width of the muscle were identified at the glabella and sellion. In the longitudinal view, the position and depth of the ICV were observed. The depth of the procerus was measured on the acquired US images using an image analysis program (Image J, National Institutes of Health, Bethesda, MD, USA).

All procedures performed in this study were approved by the institutional review board of the Yonsei University College of Dentistry (IRB identification code: No. 2-2017-0023; date of approval: 22 Jun 2017), and the subjects provided written informed consent after they had received a sufficient explanation of the study purpose and protocols.

### 5.3. Statistical Analysis

All measured values were presented as mean and standard deviation values. The unpaired Student’s t-test was used to compare the US and 3D images. The position of the ICV was analyzed using the ratio of the distance from the skin to the perpendicularly nearest wall of the ICV and the distance from the ICV to the superficial surface of the procerus. Also, the ratio of the distances to the glabella and sellion was calculated. The width, depth, and thickness of the glabella and the sellion were compared between US and 3D images using the paired t-test. The width, depth, and thickness were compared between the two types of procerus using the unpaired t-test.

## Figures and Tables

**Figure 1 toxins-11-00560-f001:**
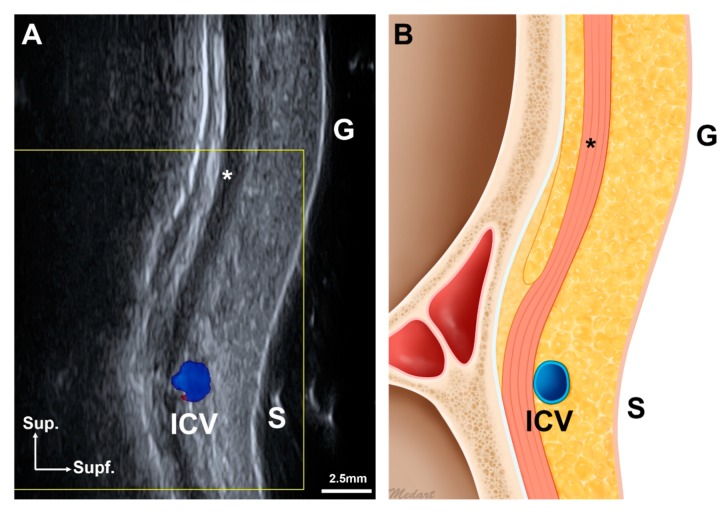
Midline longitudinal view of the ultrasonographic imaging (**A**) between the glabella and the sellion and its schematic image (**B**). Abbreviations: sup., superior; supf., superficial; G, glabella; S, sellion; ICV, intercanthal vein; asterisk, the procerus.

**Figure 2 toxins-11-00560-f002:**
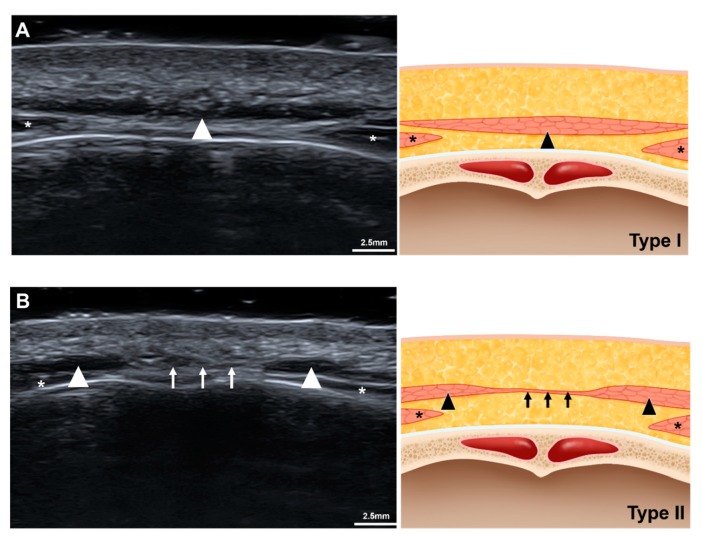
Two types of the procerus in accordance with the shape in the transverse view of the ultrasonographic imaging. (**A**) Ultrasonographic image and its illustration of type I procerus muscle. (**B**) Abbreviations: Ultrasonographic image and its illustration of type II procerus muscle. arrowhead, the procerus; asterisk, the corrugator supercilli; arrow, fibrous band between the procerus.

**Figure 3 toxins-11-00560-f003:**
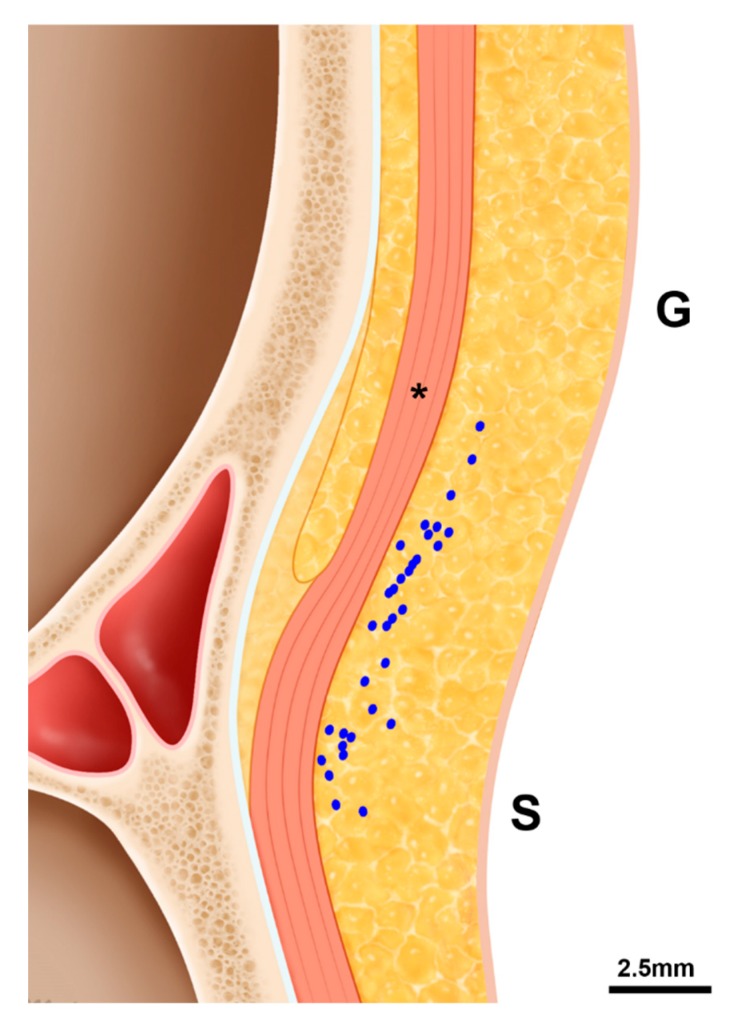
The distribution pattern of the intercanthal vein at the midline sagittal view. Abbreviations: G, glabella; S, sellion; asterisk, the procerus.

**Figure 4 toxins-11-00560-f004:**
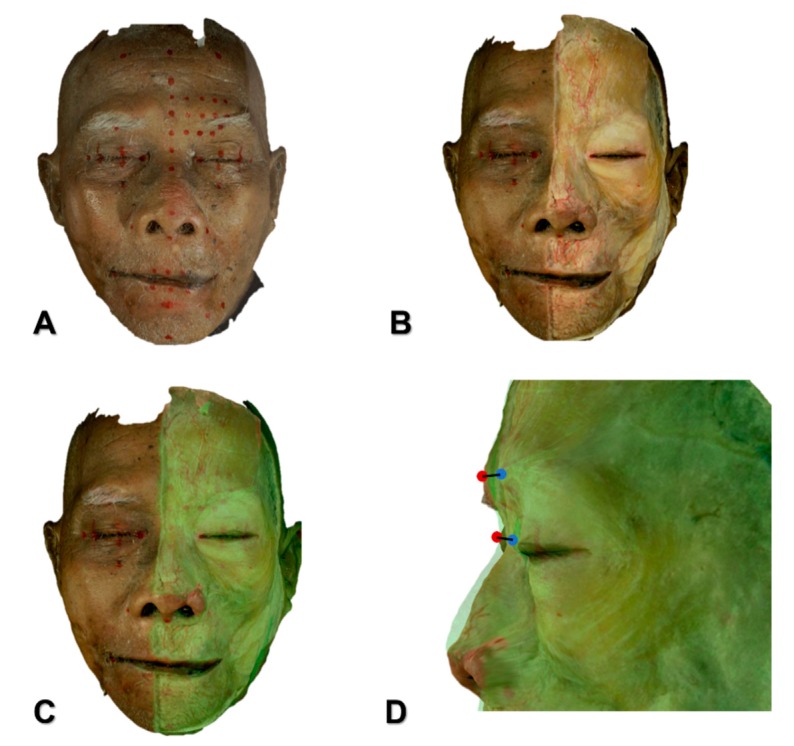
Measurement of the depth of the procerus at the glabella and sellion point using three-dimensional scanning imaging. (**A**) Before dissection, reference points were defined, then face was scanned. (**B**) The skin and subcutaneous tissue layer were removed on the left half of the face, including the midline, the face was scanned again. (**C**) The two reconstructed 3D images were superimposed based on the undissected side of the face. (**D**) Facial skin points of the glabella (red dot) and sellion were projected vertically to the points of the procerus (blue dot). The distance between the two points was calculated automatically.

**Table 1 toxins-11-00560-t001:** Measurements of the depth of the procerus muscle.

Site	Measurement Method	Depth from Skin Surface to the Procerus (mm)	*p*-Value
Glabella	Ultrasonography (*n* = 55)	3.8 ± 0.7	0.144
	3D scanning (*n* = 45)	4.1 ± 1.3	
Sellion	Ultrasonography (*n* = 69)	2.7 ± 0.6	0.063
	3D scanning (*n* = 45)	3.0 ± 1.1	

**Table 2 toxins-11-00560-t002:** Comparison of the depth, thickness, and width between the two types of procerus.

Site Measurement		Type I	Type II
		*n* = 55 (79.7%)	*n* = 14 (20.3%)
Glabella	Depth	3.8 ± 0.7 mm	
	Thickness	1.1 ± 0.5 mm	
Sellion	Depth	2.7 ± 0.5 mm	2.7 ± 0.9 mm
	Thickness *	1.6 ± 0.6 mm	1.3 ± 0.4 mm
	Width	10.1 ± 2.0 mm	10.2 ± 1.2 mm

* = *p*-value < 0.05.

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
