# Peer review of "Ultrasonographic and Three-Dimensional Analyses at the Glabella and Radix of the Nose for Botulinum Neurotoxin Injection Procedures into the Procerus Muscle"

_toxins, 2019, doi:10.3390/toxins11100560_

Round 1

Reviewer 1 Report

The authors measured depth and thickness of the procerus muscle and ICV using US imaging in healthy subjects and SD scanning images of cadavers. Both results were compared. Results can help providing anatomical landmarks for injecting BoNT in clinical use.

General impression: This manuscript provides new and principally interesting anatomical insights, useful for BoNT application into the M. procerus.

However, I have some remarks that have to be considered:

1) More general points: Correct in the whole ms:

say … botulinum neurotoxin …

use always spaces (example) … a.b ± y.z mm

Look for spelling errors.

2) Special points

Title: mention that the procerus muscle is studied

Abstract: mention that 2 methods were used for measurement

Keywords: please find other keywords that really express the important facts

Introduction:

Line 22: please rephrase so that this sentences is not exactly that used in the abstract

Line 26: The procerus is not “between” the glabella and nose, it covers the glabella, it is “at” the glabella (it is measured there by the au.)

Results

Line 47: Figure 1A or B?

Line 50f: where shall I see the difficulty?

Line 52: Figure 1A or B?

What is the meaning of the quadratic line in Fig 1A

Please put Figs 1A and B directly side by side so that one can compare them directly. Please use symbols or even better lines with demarcations (delineations)  so that the reader can have an clear impression (and control) where which measures were taken. Insert a scale bar.

In legend of Fig 1 insert (A) and (B) at the appropriate sites

Line 62: Head lie of Tab 1: cancel the word thickness – measures of thickness are not found in the Tab

Line 63: to what refers the * ?

Fig 2: Please put in Figs 2A to D original images directly side by side to the schematic diagrams so that one can compare them directly. Please use symbols or even better lines with demarcations of the layers so that the reader can have a clear impression where which measures were taken. Insert a scale bar. The arrowheads and arrows do not mark the comparable structures exactly. As the widths of the layers are the topic of the ms their exact delineations (the crucial parameter for measurement) have to be included and shown in the figs.

Line 8: please explain the word “three”

Lined 85: can be cancelled – never used in the other legends (it is mentioned in the text of the ms)

Line 92, Fig 3: what is the meaning of the Fig ? There is a wide variability of the IVC. When a frontal 3D view is used then the ICV is not a spot – it is a complete vessel. As in the conclusion the ICV takes 1/3 of the text, the results must allow that.

Fig. 3: As very exact measures are given, a very very clear and exact and reproducible definition of the glabella and sellion is crucial – however, I could not find it throughout the ms. As these definitions are extremely important for the study, please really focus on that.

Are the given measured means or medians – please insert the SD. Why are the measures in the picture others than those written in the text?

What is the important message of the fig 3? It can be given side by side with fig 4.

Discussion

Please exactly explain where the clinicians inject the Botulinumtoxin? That must be discussed to strengthen the meaning of the study. i.e. the reason why the measurements were done at the level of G and S.

Please discuss whether the results of the US measurements (young population) are a useful for the comparison with the cadaver measures (old population).

Line 120: why is the vein invisible? Why did you not measure the width of the ICV?

Line 178: why do you mention the rhinion point – was is the significance for the study ?

Line 194: I never heard of a double blind funding – please explain

Author Response

We are grateful for the very constructive advice given by the reviewers. We did our best in revising the manuscript with available data and to share the excitement we had in this study.

<Reviewer 1>

The authors measured depth and thickness of the procerus muscle and ICV using US imaging in healthy subjects and SD scanning images of cadavers. Both results were compared. Results can help providing anatomical landmarks for injecting BoNT in clinical use.

General impression: This manuscript provides new and principally interesting anatomical insights, useful for BoNT application into the M. procerus. However, I have some remarks that should be considered:

Query 1. Correct in the whole ms:

say … botulinum neurotoxin … use always spaces (example) … a.b ± y.z mm Look for spelling errors.

(A1) Thank you for the valuable advice. We revised as your advice. All changes were marked in red colored.

Query 2. Title: mention that the procerus muscle is studied

(A2) We revised the title as the below.

New title: Ultrasonographic and three-dimensional analyses at the glabella and radix for botulinum neurotoxin injection procedures into the procerus muscle

Query 3. Mention that 2 methods were used for measurement in the abstract.

(A3) We revised and added it.

The aim of this study was to measure the depth and thickness of the procerus and identify the location of the intercanthal vein using ultrasonographic (US) imaging and three-dimensional scanning method, which is needed to know to avoid side effect during BoNT injections.

Query 4. Keywords: please find other keywords that really express the important facts

(A4) Thank you for your kind advice. Reviewer 2 also advised about the keyword, and he/she suggested keywords as below. After the review among the authors, we accepted these.

Keywords: procerus; M. procerus; glabella; sellion; radix of the nose; botulinum neurotoxin injection; facial rejuvenation procedures; ultrasonographic imaging; 3D scanning imaging; glabellar transverse line.

Introduction:

Query 5. In line 22, please rephrase so that this sentence is not exactly that used in the abstract

(A5) Thank you for your advice. We revised as the below.

Botulinum neurotoxin (BoNT) injections into procerus muscle is widely used for facial rejuvenation to decrease wrinkles in cases of glabellar transverse lines or rhytids

Query 6. Line26, the procerus is not “between” the glabella and nose, it covers the glabella, it is “at” the glabella (it is measured there by the au.)

 (A6) Thank you for your advice. We revised as the below.

The procerus (M. procerus, pyramidalis nasi, depressor glabellae) is a small muscle located at the glabella and upper part of the nose. It arises from the fascia which is located near the junction of the nasal bones, and the upper lateral nasal cartilage

Results

Query 7. In line 47, clarify which Figure is marked.

(A7) Thank you for your advice. I revised Fig 1’s legend, also mentioned more clearly about Fig 1 in manuscript. And it was Figure 1A, so it was revised.

Query 8. In line 50, where shall I see the difference?

(A8) Thank you for your advice. To be shown the difference, the p-value was added in table 1.

Table 1. Measurements of the depth of the procerus muscle.

Site

Measurement method

Depth from skin surface
to the procerus (mm)

p-value

Glabella

Ultrasonography (n=55)

3.8 ± 0.7

0.144

3D scanning (n=45)

4.1 ± 1.3

Sellion

Ultrasonography (n=69)

2.7 ± 0.6

0.063

3D scanning (n=45)

3.0 ± 1.1

Query 9. In line 52, Figure 1A or B?

(A9) Thank you for your advice. It had been Fig supplementary 1, but we decided to remove supplementary Fig 1. Also, we decided to remove the results which is related to Fig S1, which is about the connection between the nasalis and procerus, however which is not related to the whole subject, botox injection into the procerus.

Query 10. What is the meaning of the quadratic line in Fig 1A?

(A10) Thank you for your advice. The quadratic line is for ROI for doppler analysis in US image for identify vessels. We discussed that we would add the explanation or remove the quadratic line in manuscript, however, both of choice does not seem appropriate. if the explanation is added, the description would be too long. If the quadratic line is removed, the figure would seem as manipulated.

Query 11. Please put Figs 1A and B directly side by side so that one can compare them directly. Please use symbols or even better lines with demarcations so that the reader can have a clear impression where which measures were taken. Insert a scale bar. In legend of Fig 1 insert (A) and (B) at the appropriate sites

(A11) We had also wanted side by side arrangement, so we arranged in revised manuscript. It will help readers for the clear impression. Also, we added a scale bar and description of Fig 1.

Query 12. In table 1

Line 62: Head lie of Tab 1: cancel the word thickness – measures of thickness are not found in the Tab Line 63: to what refers the * ?

(A12) Thank you for the valuable advice. These had been from the draft, but we changed it before the submission. However, we had missed it during the correction and approval.

Query 13. In fig 2, please put in Figs 2A to D original images directly side by side to the schematic diagrams so that one can compare them directly. Please use symbols or even better lines with demarcations of the layers so that the reader can have a clear impression where which measures were taken. Insert a scale bar. The arrowheads and arrows do not mark the comparable structures exactly. As the widths of the layers are the topic of the ms their exact delineations (the crucial parameter for measurement) have to be included and shown in the figs.

(A13) Thank you for the valuable advice. Based on your opinion (Also reviewer 2), we modified figure 2. We arranged the arrowhead, width, also we added scale bar.

Query 14. Table 2.

In line 82, please explain the word “three” Lined 85: can be cancelled – never used in the other legends (it is mentioned in the text of the ms)

(A14) Thank you for the valuable advice. These had been from the draft, but we changed it before the submission. However, we had missed it during the correction and approval.

Query 15. Fig. 3, As very exact measures are given, a very very clear and exact and reproducible definition of the glabella and sellion is crucial – however, I could not find it throughout the ms. As these definitions are extremely important for the study, please really focus on that.

(A15) Thank you for your advice. We had accepted general meaning of the glabella and sellion. As an anatomist (Most of us are anatomist), it is too general concept which is no need to be mentioned. However, it is better to clarify the definition for the reader, and it will be helpful to be reproductive. Glabella was defined as the most protruding and midline point on the smooth part of the forehead above and between the eyebrows. Sellion was defined as the deepest midline point between the forehead and nose. The same definition also applied to US analyses.

Query 16. In Fig 3,

What is the meaning of the Fig ? There is a wide variability of the IVC. When a frontal 3D view is used then the IVC is not a spot – it is a complete vessel. Are the given measured means or medians. please insert the SD. Why are the measures in the picture others than those written in the text? What is the important message of the fig 3? It can be given side by side with fig 4.

(A16) Thank you for your advice. We agreed that it was overlapped with other figures. Fig 3 is removed now. Instead, we added more information in manuscript.

Discussions

Query 17. Please explain where the clinicians inject the Botulinumtoxin? That must be discussed to strengthen the meaning of the study. i.e. the reason why the measurements were done at G and S.

(A17) Thank you for your advice. In our practice, the point of the procerus we inject is between medial ends of the eyebrow on the midline (point 1) or the point on the imaginary line from the 45 degrees from the medial ends of the eyebrow (under the point 1). We tried to find literature reference, most of the references are mentioned ‘depends on the physicians’. Description about the injection procedure would reinforce what we want to state, however without the literature reference, we doubt about the mention about the injection point in this manuscript is proper or not.

Query 18. Please discuss whether the results of the US measurements (young population) are a useful for the comparison with the cadaver measures (old population).

(A18) Thank you for your advice. We mentioned the reason why we used both modality on line 130-138. Yes. As you mentioned, face anatomy is quite different between the old and young. Therefore, using both modalities is the one of strong point of this research, we think.

Query 19. In line 120, why is the vein invisible? Why did you not measure the width of the ICV?

(A19) Thank you for your advice. After the review among the authors, the contents about it is added as below.

This difference is likely due to the limited resolution of US imaging device, the pressure during US scanning, and reduced blood flow of ICV in some subjects who received US scanning.

Query 20. In line 178, why do you mention the rhinion point – was is the significance for that ?

(A20) Thank you for your advice. The rhinion point had been discussed in supplementary material, but now it was removed.

Query 21. In line 194, I never heard of a double blind funding – please explain

(Q21) It is because of the policy of toxins journal policy, however we clearly stated about funding in manuscript from one research foundation.

Query 22. In line 196, the list is not as requested by the journal

(Q22) Thank you for your advice. We checked it again, but we think reference style is matched with this journal.

Reviewer 2 Report

The authors determined the depth and thickness of the procerus and ICV using US imaging in healthy subjects and using SD scanning images of cadavers to provide clinical anatomical information when injecting BoNT at the glabella and radix of the nose.

This manuscript provides the new interesting anatomical findings for use when injecting BoNT into the M. procerus.

Indeed, it should be corrected before publishing:

Generally (in all manuscript):

Please use instead of “botulinum toxin”, “ botulinum neurotoxin” term Please add into the introduction and into the discussion more about the clinical relevance of our results Please check and correct all spelling errors in the manuscript Please check and correct all spaces between the ciphers and mathematics symbols. Example:

           instead of 3.8±0.7 mm, please change to  3.8 ± 0.7 mm.

What do you mean with “point”: (p=.006? or p<.001)? Please you the common designation.

Specially:

Abstract

Line 7 please add also another term: ..the procerus muscle (M. procerus).

Please check and correct all spaces between the ciphers and mathematic symbols.

Keywords

Please use these keywords: procerus; M. procerus; glabella; sellion; radix of the nose; botulinum neurotoxin injection; facial rejuvenation procedures; ultrasonographic imaging; 3D scanning imaging; glabellar transverse line.

Introduction

Line 23: Also, Please describe in the introduction the exactly point for BoNT injections in the clinical praxis.

Line 26: please correct and add also the references at the end of the sentence: The procerus (M. procerus, pyramidalis nasi, depressor glabellae) is a small mimic muscle located between the glabella and nose. This is not correct to writing that the procerus located between the glabella and nose!

Line 26-33: Please note the right exactly position, more information about the M. procerus as well as the appropriate references. Also, it is very important to described the fascia of the M. procerus.

What about the fat compartment in this area?

Line 28: please correct: please use instead of “radix”, “radix of the nose” and add the references at the end of the sentence.

Line 30: except the innervation, what about the Blood vessels for M. procerus?

Line 33: do you know some another “danger” structures except the ICV in this zone?

Line 38: please add more anatomical information and variability of ICV.

Results

Line 47, 52: please add corrects to Figure 1: A or B?

Figure 1: please change to Figure 1 A,B and describe very exactly what is “A” and what is  “B” images separately. Also is very important always to note:  Is it the US imaging? Is it  the 3D imaging?  Is it the schematic image? Please put both images side by side.  

Figure 2: please change to Figure 2 A-D and describe very exactly “A” to “D” images very exactly separately. Also is very important always to note: Is it the US imaging? Is it  the 3D imaging?  Is it the schematic image?

Please delete the “dot” between the .. imaging and arrowhead..

Line 86: please correct “longitudinal-view” to “longitudinal view”.

Figure 3: Is it one schematic image or is it one 3D scanning image?  Please describe the Figure very exactly.

Figure 4: describe it very exactly, the same with Figure 1, 2 and 3! Which kind of image?    

Table 1: what do you mean with “*”?

Table 2: what do you mean with line 85?

Discussion

Line 98-103: I’m wondering that you compare in your study the US images of the young healthy subjects and the SD scanning images of the old cadavers? Please explain why and if it possible, because, the face anatomy of the old persons is quite different compare with young subjects?

Line 102: please correct to .. radix of the nose.

Line 108: please correct to .. “injection site” instead of “injection-site”

Figure5: Please note exactly the method! Is it one 3D scanning image or another one?

Line 171: please change “noninvasive” to “non-invasive”.

 Line 178: why do you discuss the rhinion point now and never before in your ms?

Author Response

We are grateful for the very constructive advice given by the reviewers. We did our best in revising the manuscript with available data and to share the excitement we had in this study.

Reviewer 2.

The authors determined the depth and thickness of the procerus and ICV using US imaging in healthy subjects and using SD scanning images of cadavers to provide clinical anatomical information when injecting BoNT at the glabella and radix of the nose.

This manuscript provides the new interesting anatomical findings for use when injecting BoNT into the M. procerus. Indeed, it should be corrected before publishing:

Generally (in all manuscript):

Query 1.

Please use instead of “botulinum toxin”, “ botulinum neurotoxin” term Please check and correct all spelling errors in the manuscript Please check and correct all spaces between the ciphers and mathematics symbols. Example please correct: please use instead of “radix”, “radix of the nose” and add the references at the end of the sentence.

(A1) Thank you for the valuable advice. We revised as your advice. All changes were marked in red colored.

Query 2. Please add into the introduction and into the discussion more about the clinical relevance of our results

(A2) Thank you for your advice. There were lots of published articles about the adverse effect why we should do this kind of research. We also mentioned these articles more in discussion part, line 105-114. We tried to add introduction with the detail also, however if so, it looks too overlapped.

Abstract

Query 3. In line 7, please add also another term: ..the procerus muscle (M. procerus).

(A3) Thank you for your advice. We revised in introduction part due to word limit in abstract.

Keywords

Query 4. Please use these keywords: procerus; M. procerus; glabella; sellion; radix of the nose; botulinum neurotoxin injection; facial rejuvenation procedures; ultrasonographic imaging; 3D scanning imaging; glabellar transverse line.

(A4) Thank you for your valuable advice. After the review among the authors, we accepted whole your advice about the keywords.

Introduction

Query 5. In line 23, Also, Please describe in the introduction the exactly point for BoNT injections in the clinical praxis.

(A5) Thank you for your advice. In our practice, the point of the procerus we inject is between medial ends of the eyebrow on the midline (point 1) or the point on the imaginary line from the 45 degrees from the medial ends of the eyebrow (under the point 1). We tried to find literature reference, most of the references are mentioned ‘depends on the physicians’. Description about the injection procedure would reinforce what we want to state, however without the literature reference, we doubt about the mention about the injection point in this manuscript is proper or not.

Query 6.

In line 26, please correct and add also the references at the end of the sentence: The procerus (M. procerus, pyramidalis nasi, depressor glabellae) is a small mimic muscle located between the glabella and nose. This is not correct to writing that the procerus located between the glabella and nose! Line 26-33: Please note the right exactly position, more information about the M. procerus as well as the appropriate references. Also, it is very important to described the fascia of the M. procerus. What about the fat compartment in this area? except the innervation, what about the Blood vessels for M. procerus?

(A6) Thank you for your advice. We added contents in accordance with your advice, also we changed our expressions about ‘between the glabella and nose!

About the fat compartment, subprocerus galeal fat is located deep in this area, which is demarcated by dense connective tissues. However, we are not sure that our result is not quite relevant of it. And the blood supply is from branches of the facial artery, we added it in manuscript.

Query 7. In line 33, do you know some another “danger” structures except the ICV in this zone?

(A7) Thank you for your advice. Dorsal nasal artery, supratrochleal vessels and other adjacent muscles might be danger structures during BoNT injections into the procerus. However, to consider where the BoNT was injected into the procerus (mostly midline point), It is not quite relevant with our manuscript here.

Query 8. In line 38. please add more anatomical information and variability of ICV.

 (A8) Thank you for your advice. Some of our authors performed cadaveric studies before, also we mentioned it in the manuscripts. However we added more information.

In the radix of the nose, ICV communicated with and connected angular veins bilaterally.

Results

Query 9. Figure 1.

Line 47, 52: please add corrects to Figure 1: A or B? Figure 1: please change to Figure 1 A,B and describe very exactly what is “A” and what is “B” images separately. Also is very important always to note:  Is it the US imaging? Is it  the 3D imaging?  Is it the schematic image? Please put both images side by side.

(A9) Thank you for your advice. In fig 1,line 52 was about Fig 1S. However, we decided to remove Fig 1S which is not relevant with BoNT injection into the procerus. We modified the manuscript and figure legend about the figure 1 properly.

Midline longitudinal view of the ultrasonographic imaging (A) between the glabella and the sellion and its schematic image (B). sup., superior; supf., superficial; G, glabella; S, sellion; ICV, intercanthal vein; asterisk, the procerus.

Query 10. Figure 2, please change to Figure 2 A-D and describe very exactly “A” to “D” images very exactly separately. Also is very important always to note: Is it the US imaging? Is it the 3D imaging?  Is it the schematic image? Please delete the “dot” between the .. imaging and arrowhead..

(A10) Thank you for the advice. For the better explanation and expression, we modified figure 2 as your mention – side by side arrangement. Also, we changed figure legend as below.

Two types of the procerus in accordance with the shape in the transverse view of the ultrasonographic imaging. (A) Ultrasonographic image and its illustration of type I procerus muscle. (B) Ultrasonographic image and its illustration of type II procerus muscle. arrowhead, the procerus; asterisk, the corrugator supercilli; arrow, fibrous band between the procerus.

Query 11. In line 86, please correct “longitudinal-view” to “longitudinal view”.

(A11) Thank you for your advice. we revised.

Query 12. In figure 3, Is it one schematic image or is it one 3D scanning image?  Please describe the Figure very exactly. Also in figure 4, describe it very exactly, the same with Figure 1, 2 and 3! Which kind of image?   

(A12) Thank you for your advice. Fig 3 and 4 had been overlapped, so Fig 3 was removed now, as other reviewer’s comment. However, we reinforced the manuscript what we try to be mentioned in Fig 3. Because we removed overlapped contents, Figure 4 became more meaningful now.

Query 13. Table 1: what do you mean with “*”?

(A13) Thank you for the valuable advice. These had been from the draft, but we changed it before the submission. However, we had missed it during the correction and approval. Thank you very much!

Query 14. Table 2: what do you mean with line 85?

(A14) Thank you for the valuable advice. These had been from the draft, but we changed it before the submission. However, we had missed it during the correction and approval. Thank you very much!

Discussion

Query 15. In line 98-103, I’m wondering that you compare in your study the US images of the young healthy subjects and the SD scanning images of the old cadavers? Please explain why and if it possible, because, the face anatomy of the old persons is quite different compare with young subjects?

(A15) Thank you for your advice. We mentioned the reason why we used both modality on line 130-138. Yes. As you mentioned, face anatomy is quite different between the old and young. Therefore, using both modalities is the one of strong point of this research, we think.

Query 16. please correct to .. “injection site” instead of “injection-site”

(A16) Thank you for your advice. we revised.

Query 17. In figure 5, Please note exactly the method! Is it one 3D scanning image or another one?

(A17) Thank you for your advice. We clarified the legend as below.

Measurement of the depth of the procerus at the glabella and sellion point using three-dimensional scanning method. Facial skin points of the glabella (red dot) and sellion is projected vertically to the points of the procerus (blue dot). The distance between two points was calculated automatically.

Query 18. In line 171, please change “noninvasive” to “non-invasive”.

(A18) Thank you for your advice. we revised.

 Query 19. In line 178, why do you discuss the rhinion point now and never before in your ms?

(A19) Thank you for your advice. US images at the rhinion point had been discussed in the supplementary material. However, supplementary material was removed.

Round 2

Reviewer 1 Report

The ms has massively and adaquately improved.

The editor should reduce the size of Fig 3 to the size as shown in  Fig 1b for better comparison.

Author Response

 We are grateful for the very constructive advice given by the reviewers. We did our best in revising the manuscript with available data and to share the excitement we had in this study.

Query 1. The editor should reduce the size of Fig 3 to the size as shown in Fig 1b for better comparison.

 (A1) We changed the size of Fig. 3 as same size in Fig 1b.

Reviewer 2 Report

Dear authors,

thank you very much for all corrections. But; I still have some proposals which should be done before publishing:

Title: please use "radix of the nose" 

Introduction:

In the radix of the nose ICV communicated with and connected angular veins bilaterally. Please add the references at the end of this sentence. Also, please add some information and references about connection of angular vein to the cavernous sinus and cavernous sinus thrombosis as well as significance for the clinical practice.   

Results:

Figure 1: Please add all  apaces between the numbers and characters: n = 55; n = 45 and so on..

Don't start your sentence with sup., superior; supf., superficial; G, glablellaglabella; S, sellion; ICV, intercanthal vein; asterisk, the procerus.

Please say after "B": Abbreviations:sup., superior; supf., superficial; G, glablellaglabella; S, sellion; ICV, intercanthal vein; asterisk, the procerus.

Figure 2: Please say: Abbreviations: arrowhead, the procerus; asterisk, the corrugator supercilli; arrow, fibrous band between the procerus. Please use:*p < 0.05.

Figure 3: Please say: Abbreviations: S, sellion; asterisk, the procerus.

Table 2: Please use:*p < 0.05.

Discussion:

Please add into Discussion about the common points for BoNT-A injections in the clinical praxis of the procerus. Please add references and discuss also, even if written, as you mentioned, "depends on the physicians".   

Please add more references at the end of the sentence: We compared the depth of the procerus between US images and 3D-scanning images of cadavers. Because these two methods have been validated previously [121]

Author Response

 We are grateful for the very constructive advice given by the reviewers. We did our best in revising the manuscript with available data and to share the excitement we had in this study.

Query 1. Title: please use "radix of the nose"

(A1) Thank you for your advice. we revised.

Introduction

Query 2. In the radix of the nose ICV communicated with and connected angular veins bilaterally. Please add the references at the end of this sentence. Also, please add some information and references about connection of angular vein to the cavernous sinus and cavernous sinus thrombosis as well as significance for the clinical practice.  

(A2) Thank you for your advice. Your opinion reinforced why ICV is important. We added as your comment.

Results

Query 3. In Figure 1, Please add al spaces between the numbers and characters: n = 55; n = 45 and so on..

(A3) Thank you for your advice. We had missed it even you had given an advice. Sorry for our inattention. We revised.

Query 4. Don't start your sentence with sup., superior; supf., superficial; G, glablellaglabella; S, sellion; ICV, intercanthal vein; asterisk, the procerus.

Please say after "B": Abbreviations:sup., superior; supf., superficial; G, glablellaglabella; S, sellion; ICV, intercanthal vein; asterisk, the procerus.

Figure 2: Please say: Abbreviations: arrowhead, the procerus; asterisk, the corrugator supercilli; arrow, fibrous band between the procerus. Please use:*p < 0.05.

Figure 3: Please say: Abbreviations: S, sellion; asterisk, the procerus.

(A4) Thank you for your advice. we revised it as you had adviced.

Query 5. In Table 2, Please use:*p < 0.05.

(A5) Thank you for your advice. we revised it as you had adviced.

Discussion

Query 6. Please add into Discussion about the common points for BoNT-A injections in the clinical praxis of the procerus. Please add references and discuss also, even if written, as you mentioned, "depends on the physicians".  

(A7) Also it depends on case by case, even it is different between the ethnics. For the better understanding and for the clinical guide for the BoNT injections into the procerus, we added the information about the common points into the procerus.

Query 7. Please add more references at the end of the sentence: We compared the depth of the procerus between US images and 3D-scanning images of cadavers. Because these two methods have been validated previously [121]

(A7) Thank you for your advice. Our research group used two modalities at the first, now we are now doing lots of research using both modalities, and it validated several times. As your opinion, we added more references about it.